# Smartphone screen testing, a novel pre-diagnostic method to identify SARS-CoV-2 infectious individuals

Rodrigo M Young[1,2]*, Camila J Solis[1], Andres Barriga-Fehrman[1], Carlos Abogabir[1], Alvaro R Thadani[1], Mariana Labarca[1], Eva Bustamante[3], Cecilia V Tapia[4], Antonia G Sarda[4], Francisca Sepulveda[4], Nadia Pozas[4], Leslie C Cerpa[5], María A Lavanderos[5], Nelson M Varela[5], Alvaro Santibañez[6,7], Ana M Sandino[6,8], Felipe Reyes-Lopez[6,8,9,10], Garth Dixon[9,11,12], Luis A Quiñones[5]*

[1]Diagnosis Biotech, Santiago, Chile; [2]Institute of Ophthalmology, University College London, London, United Kingdom; [3]Grupo Oncológico Cooperativo Chileno de Investigación, Santiago, Chile; [4]Laboratorio de Especialidad, Clínica Dávila, Santiago, Chile; [5]Laboratory of Chemical Carcinogenesis and Pharmacogenetics (Chilean Health Council Diagnostic Covid19 Laboratory), Department of Basic and Clinical Oncology, Faculty of Medicine, University of Chile, Santiago, Chile; [6]Centro de Biotecnología Acuícola, Facultad de Química y Biología, Santiago, Chile; [7]Programa Disciplinario de Inmunología, Instituto de Ciencias Biomédicas, Facultad de Medicina, Universidad de Chile, Santiago, Chile; [8]Departamento de Biología, Facultad de Química y Biología, Universidad de Santiago de Chile, Santiago, Chile; [9]Department of Cell Biology, Physiology and Immunology, Universitat Autònoma de Barcelona, Bellaterra, Spain; [10]Facultad de Medicina Veterinaria y Agronomía, Universidad de Las Américas, Providencia, Chile; [11]Department of Microbiology, Virology and Infection Control, Hospital National Health Service Foundation Trust, London, United Kingdom; [12]National Institute for Health Research Biomedical Research Centre at Great Ormond Hospital for Children National Health Service Foundation Trust and University College, London, United Kingdom

*For correspondence:
ryoung@diagnosis-bt.com;
rodrigo.young@ucl.ac.uk (RMY);
lquinone@uchile.cl (LAQ)

**Abstract** The COVID-19 pandemic will likely take years to control globally, and constant epidemic surveillance will be required to limit the spread of SARS-CoV-2, especially considering the emergence of new variants that could hamper the effect of vaccination efforts. We developed a simple and robust – *Phone Screen Testing* (*PoST*) – method to detect SARS-CoV-2-positive individuals by RT-PCR testing of smartphone screen swab samples. We show that 81.3–100% of individuals with high-viral-load SARS-CoV-2 nasopharyngeal-positive samples also test positive for *PoST*, suggesting this method is effective in identifying COVID-19 contagious individuals. Furthermore, we successfully identified polymorphisms associated with SARS-CoV-2 Alpha, Beta, and Gamma variants, in SARS-CoV-2-positive *PoST* samples. Overall, we report that *PoST* is a new non-invasive, cost-effective, and easy-to-implement smartphone-based smart alternative for SARS-CoV-2 testing, which could help to contain COVID-19 outbreaks and identification of variants of concern in the years to come.

## Introduction

The coronavirus disease 2019 (COVID-19) pandemic, caused by the SARS-CoV-2 virus, has had a massive impact on human lives, health, and quality of life, challenging countries' healthcare systems

and their economies worldwide. Several countries are now facing a second or third wave of COVID-19 cases. Moreover, the development of this pandemic is still difficult to predict considering the emergence of SARS-CoV-2 mutations located in the receptor-binding domain of the surface spike protein, which creates a new hazard as evolved viral strains may be more infectious or evade the immune response and hinder vaccination efforts (*CDC, 2021a*; *Collier et al., 2021*; *Greaney et al., 2021a*; *Volz et al., 2021*; *Wang et al., 2021*; *Wibmer et al., 2021*). Particularly, the more infectious Alpha variant (B.1.1.7, *Frampton et al., 2021*; *Rambaut et al., 2020*), together with the Beta variant (B.1.351, *Tegally et al., 2020*), Delta (B.1617.2, *Torjesen, 2021*) and Gamma variant (P.1, *Faria et al., 2021* ), considered variants of concern by CDC, which share the antigenic drift and/or signature mutations E484K, L452R,K417N and N501Y in the region of the spike protein that is recognised by neutralising antibodies (*Collier et al., 2021*; *Tchesnokova et al., 2021*; *Greaney et al., 2021b*; *Wang et al., 2021*; *Wibmer et al., 2021*).

Early in the pandemic, the RNA sequence of the SARS-CoV-2 virus was made available (*Wu et al., 2020*), enabling the testing of infected patients by reverse transcription (RT)–PCR (*Arnaout et al., 2020*; *Corman et al., 2020*). Regular and broad testing of SARS-CoV-2 seems essential to contain the propagation of SARS-CoV-2, as many infected individuals express no symptoms, inadvertently spreading the infection (*Ferretti et al., 2020*; *Kronbichler et al., 2020*; *Petersen and Phillips, 2020*; *Pollock and Lancaster, 2020*; *Sayampanathan et al., 2021*). Therefore, successful epidemiological surveillance strategies required to monitor the spread of SARS-CoV-2 and the outbreak of new strains should include large-scale screening methods that enable periodic and continuous testing of the general population.

However, testing capacity has been limited, hampering attempts to control the spread of SARS-CoV-2. One obstacle is that reliable nasopharyngeal sampling and RT-PCR testing is highly invasive and requires both specialised staff and appropriate conditions for the manipulation and transport of the samples to comply with clinical standards and protocols expected by regulatory bodies (*CDC, 2021b*; *UK-Government U, 2021*). Lateral flow device antigen tests are cheaper, accurate when detecting individuals with high viral load, and an epidemiologically effective option to identify SARS-CoV-2 contagious people (*Dinnes et al., 2021*; *Pavelka et al., 2021*; *Wagenhäuser et al., 2021*). Yet, correct testing also requires nasopharyngeal sampling. Therefore, regular large-scale testing is difficult because accurate tests are either too invasive, expensive, or logistically complicated to implement, which make them unviable for the task.

To provide a simple alternative to identify SARS-CoV-2-infected individuals, we designed and validated a method by which SARS-CoV-2 RNA can be RT-PCR detected from samples taken from a person's smartphone screen. This Phone Screen Testing (*PoST*) method shows high sensitivity (81–100%) compared to nasopharyngeal RT-PCR SARS-CoV-2 test results from individuals with a high viral load, making the smartphone screen a good proxy of the health status of contagious individuals. Furthermore, the SARS-CoV-2 variants present in *PoST* samples with low RT-PCR Ct value were successfully identified. Overall, our results provide a new, non-invasive, cost-effective, and efficient method to identify COVID-19 contagious infected cases and limit the transmission of the disease.

## Results

### *PoST* pilot to identify COVID-19-positive cases

SARS-CoV-2 RNA can be detected from many different kinds of surfaces, places, and devices, including phones (*Marshall et al., 2020*; *Santarpia et al., 2020*; *Ye et al., 2020*; *Zhou et al., 2020*). Furthermore, active SARS-CoV-2 virus is more likely to be recovered from some surfaces when expelled from individuals with seemingly high viral load (*Bullard et al., 2020*; *Jefferson et al., 2020*; *Sonnleitner et al., 2021*). Smartphones are personal objects that are constantly exposed to peoples' mouths, their screens becoming a likely contaminated surface. Therefore, we hypothesised that COVID-19 contagious individuals will regularly deposit aerosols, droplets of saliva, or upper respiratory tract secretions containing shed SARS-CoV-2 virions, over the screen of their phone, which could then be sampled and detected by RT-PCR.

To facilitate test-trace and isolating strategies of COVID-19-infected individuals, we assessed the RT-PCR detection of SARS-CoV-2 RNA from smartphone screen swab samples via *PoST*. The presence of SARS-CoV-2 RNA in phone screens was then correlated with clinical COVID-19

nasopharyngeal RT-PCR-positive diagnosis of the phones' owner. The cohort sample in this pilot study consisted of both symptomatic and asymptomatic or pre-symptomatic individuals who required SARS-CoV-2 testing due to a suspected SARS-CoV-2 infection, close contact to SARS-CoV-2-positive individuals, or return to work and travel authorisation requests in Santiago, Chile, between September and October 2020 (3.81% ± 1.0 average positivity rate in Santiago, *Figure 1—figure supplement 1C–E*).

Polyester swabs embedded in saline solution were used by a member of our team to sweep the bottom half of smartphone screens of individuals (*Figure 1A*). Of 540 individuals, 51 tested positive for SARS-CoV-2 by nasopharyngeal RT-PCR detection (9.4% positivity rate, *Figure 1B* [whole bar], *Figure 1—figure supplement 1A*, *Source data 1*). Remarkably, all samples with low Ct value, under 20, also tested positive for *PoST* (n = 15, *Figure 1B* [blue segment of bars], *Figure 1C*, *Source data 1*). Our results suggest that the ability for *PoST* to correctly identify positive nasopharyngeal individuals, sensitivity, is 100% in individuals with high viral load, Ct value under 20. As shown in *Figure 1C* (red line, *Source data 1*), the accumulated sensitivity of *PoST* in nasopharyngeal samples with medium Ct values below 30 is 89.7% (n = 29, medium and low viral load). Therefore, our results suggest that testing samples from smartphone screens is effective in identifying COVID-19-positive patients with low and medium Ct RT-PCR result values, which are thought to be actively contagious and expelling virus particles (*Bullard et al., 2020*; *Jefferson et al., 2020*; *Sonnleitner et al., 2021*). The study was performed in double-blind conditions such that the *PoST* and the nasopharyngeal RT-PCR tests were carried out in different laboratories by independent teams, which were not aware of each other's result outcome.

Of all the *PoST* and nasopharyngeal-positive samples in this cohort, 76% (n = 22/29) corresponded to individuals with no specific COVID-19 symptoms (*Figure 1—figure supplement 1E*, *Source data 1*). Therefore, testing smartphone screens is effective in identifying COVID-19-infected individuals regardless of their symptoms at the time of testing.

The overall ability of *PoST* to correctly identify a negative nasopharyngeal test, specificity, in this pilot was 98.8% (*Figure 1—figure supplement 1A*, *Source data 1*). Of this cohort, six samples were identified as *PoST*-positive/nasopharyngeal-negative, which could be interpreted as *PoST* false positives (*Figure 1—figure supplement 1*, *Source data 1*). However, in two cases that were contacted after testing, the individuals had three clear COVID-19 symptoms (*Source data 1*), suggesting that rather than a *PoST* false positive, these cases could be a nasopharyngeal false-negative result.

To further evaluate the apparent bimodal distribution observed in the Ct values from this cohort of positive COVID-19 clinical samples, we analysed the RT-PCR Ct dataset results of SARS-CoV-2-positive cases from the Davila Clinic (Santiago, Chile) adding up to over seven thousand samples (*Figure 1—figure supplement 2A-H*, *Source data 3*). When this data set was grouped in month intervals, we did observe a specific distribution of Ct values consistent with the pilot data at the months this cohort was tested (*Figure 1B*, *Figure 1—figure supplement 2G-H*).

The results of our pilot study were encouraging and suggested that the *PoST* method could be a good alternative to more invasive tests with similar sensitivity, like lateral flow antigen test screening (*Dinnes et al., 2021*; *Jääskeläinen et al., 2021*; *Wagenhäuser et al., 2021*). Therefore, validation with higher sample number was required to generate more conclusive evidence.

## Validation of phone screen testing in a high-positivity-rate cohort

To further validate the *PoST* method, we increased the sample size by testing a new set of 764 individuals at the same clinic and similar cohort kind as before, between the 5 and 16April 2021. At this time, Santiago presented a high positivity rate (12.4% ± 2.2, *Chilean Department of Health, 2021*) as a consequence of the second COVID-19 wave in the country, which enabled identifying 182 SARS-CoV-2-positive individuals (24% positivity rate in this cohort at the clinic), by nasopharyngeal swabbing (*Figure 1D*, *Figure 1—figure supplement 1B*). The distribution of RT-PCR results in this study is unimodal and tends towards lower Ct values compared to the distribution observed in the pilot study (*Figure 1B,C*).

Similar to what was observed in the pilot cohort, the sensitivity of *PoST* ranges between 81.3% and 100% in nasopharyngeal samples with low Ct value, under 20 (high viral load, *Figure 1D,E*), and the sensitivity decays at higher Ct values (*Figure 1E*, red line). In this cohort, 35% of *PoST*- and nasopharyngeal-positive samples corresponded to COVID-19 asymptomatic or pre-symptomatic

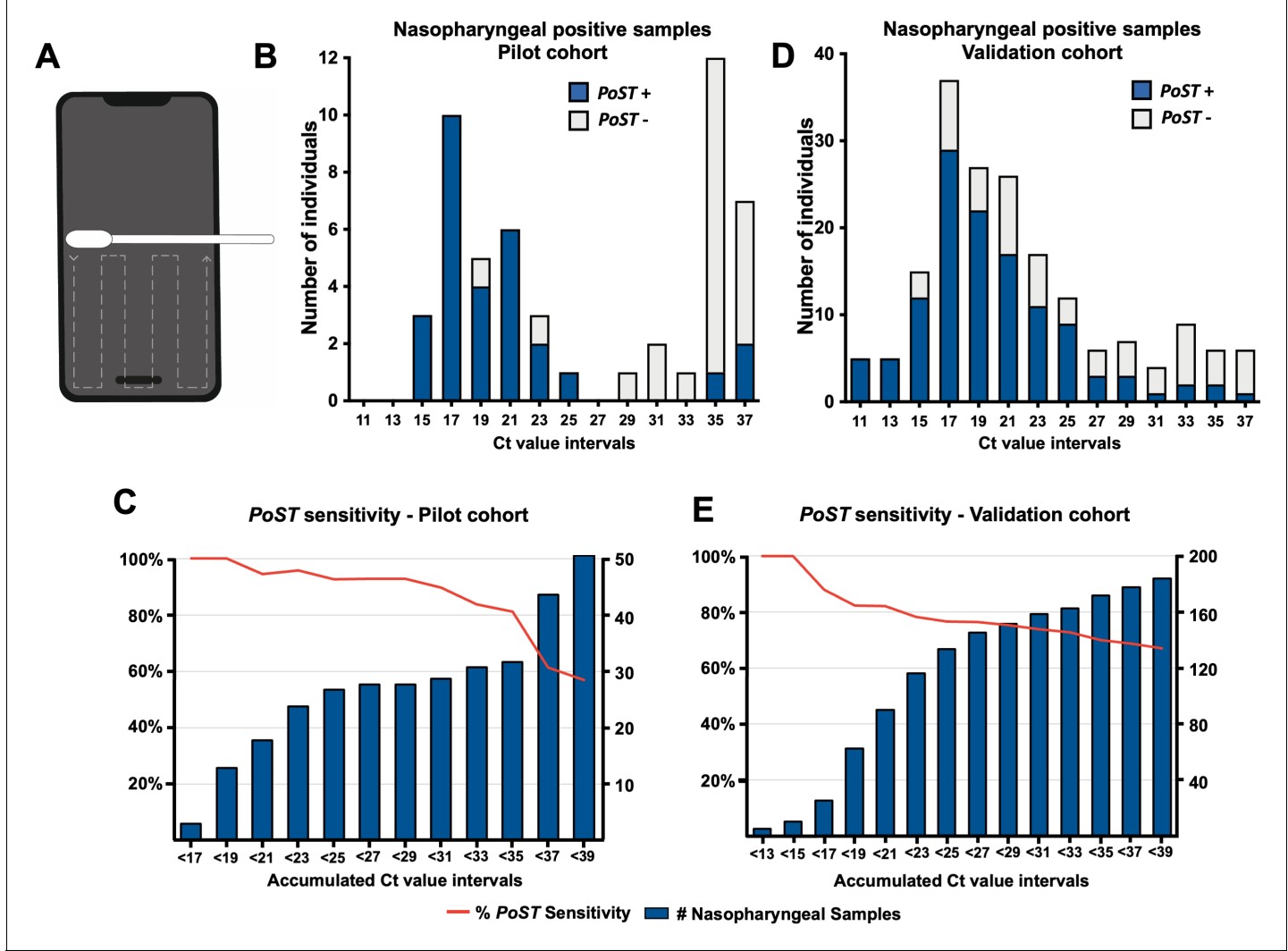

**Figure 1.** Phone Screen Testing and RT-PCR can identify individuals with high SARS-CoV-2 viral load. (A) Schematic of smartphone sampling. Swabbing follows the dashed line indicated trajectory. (B, D) Histograms showing the distribution of individuals on ranges of clinical RT-PCR Ct value results for the pilot cohort (B, full dataset in *Source data 1*) and validation cohort (D, full dataset in *Source data 2*). For example, bar Ct value 11 corresponds to samples with Ct between [11–13 [interval]]. Whole bar, all individuals with positive nasopharyngeal RT-PCR result. Blue bar section, those individuals also *PoST* positive. (C, E) Bars represent the accumulated number of individuals with positive RT-PCR clinical results under the Ct value number associated with the bar for, (C) pilot cohort, and (E) validation cohort. Red line in both plots depicts the sensitivity observed relative to clinical nasopharyngeal RT-PCR testing at the corresponding accumulated Ct value results. Left axis, percentage of sensitivity; right axis, number of accumulated individuals under the corresponding Ct value.

The online version of this article includes the following figure supplement(s) for figure 1:

**Figure supplement 1.** Pilot and validation study cohort data analysis.

**Figure supplement 2.** SARS-CoV2-19 RT-PCR cycle threshold (Ct) values in Chilean individuals.

individuals at the time of testing (n = 43/122, *Source data 2*). This validation study was also performed in the same double-blind conditions as the pilot study described above.

The overall specificity of *PoST* during this validation was 97.6% (*Figure 1—figure supplement 1B*); 14 SARS-CoV-2-negative nasopharyngeal samples were identified positively by *PoST* (*Figure 1—figure supplement 1B*, *Source data 2*). Of these cases, we could contact four individuals, of which three had clear COVID-19 symptoms, one of which tested positive when the nasopharyngeal sample was repeated (*Source data 2*). This suggests that some of the discrepancies may be due to nasopharyngeal false-negative test results.

Early in the pandemic, the notion that SARS-CoV-2 could spread via surfaces made regular disinfection of surfaces a common practice (*Goldman, 2020*). We therefore evaluated whether cleaning smartphone screens before *PoST* sampling could affect the test results. Of all the SARS-CoV-2 *PoST*-and nasopharyngeal-positive samples where the screen had been cleaned within 24 hr (n = 23), 22% (5/23) were taken from phones that had been sanitised the same day (<6 hr), and 48% (8/23) less than 2 hr before the *PoST* sample was taken (*Source data 3*). This suggests that sanitising or cleaning the smartphone before the *PoST* sampling may not affect the detection of SARS-CoV-2 traces on the phone screen.

This validation study confirms that the *PoST* method shows a high sensitivity when identifying SARS-CoV-2-infected individuals with a low RT-PCR Ct value, regardless of their symptoms, providing a new valuable tool to tackle the COVID-19 pandemic.

### SARS-CoV-2 variant identification in *PoST* samples

The current challenge in COVID-19 diagnostics is to find ways to distinguish the specific SARS-CoV-2 variant present in the tested samples to aid in containing outbreaks of variants with higher virulence or infectivity. To aid this task, we performed a screen to address whether SARS-CoV-2 variants can be identified from *PoST* samples. Fluorescent probe-based multiplexed or single RT-PCR reactions in 69 *PoST* samples from the 2021 cohort was used to identify the following variations in the Spike (S) SARS-CoV-2 gene: del69/70, K417N/T, E484K, N501Y, P681H. From this analysis, 36 samples fulfilled the criteria of amplifying SARS-CoV-2 control genes S and N. Out of this group, seven samples tested positive for at least one SNP associated with new SARS-CoV-2 variants. Two samples had all three variations described to be present in Beta (B.1.351-South African) or Gamma (P.1-Brasilian) variants (K417T, E484K, N501Y). Another three samples showed variations E484K and N501, which are only found together in Gamma and Beta variants. Even though these samples did not test positive for K417T, it is still likely that the virus in these samples corresponds to either Beta or Gamma, as the combination of variations that tested positive have only been found together in these two SARS-CoV-2 variants. One sample only tested positive for SNP K417T, which is only found in Gamma and Beta SARS-CoV-2 variants. Lastly, we identified one sample with variation del69/70 and P681H, which are only found together in the Alpha (B.1.1.7- UK) SARS-CoV-2 variant. The remaining 28 samples showed no variation, and hence are likely to contain original Wuhan SARS-CoV-2.

Our results imply that the capacity to identify SARS-CoV-2 RNA by the primer/probe set we used for *PoST* is not affected by the mutations described for the variant of concern we identified. The capacity of identifying SARS-CoV-2 variants from *PoST* samples differentiates this assay from lateral flow device antigen testing, which can only identify the presence of the virus and not discriminate specific variants.

## Discussion

A recent study suggests that socioeconomic status and delay in testing for SARS-CoV-2 was a significant factor in the high mortality rate observed during 2020 in Santiago, Chile (*Mena et al., 2021*). Therefore, finding new methods to enhance epidemiological surveillance strategies are necessary to limit the struggle generated by the COVID-19 pandemic.

Our study suggests that the *PoST* method could be an effective non-invasive way to rapidly identify COVID-19-positive contagious individuals who actively spread the virus, regardless of their symptoms. This method does not require specialised operators or conditions for sampling. Furthermore, because *PoST* is an environmental test, the protocols and reagents required to manipulate and process the samples for RT-PCR have been optimised to make *PoST* substantially more cost-effective, circumventing the need for clinical-grade reagents and standards. All which makes *PoST* a good alternative for large-scale population testing.

### Interpretation of the *PoST* method results

*PoST*-positive tests correlate with nasopharyngeal samples with low RT-PCR Ct values, which suggests that individuals with low Ct clinical RT-PCR results are probably passing through the peak of a COVID-19 infection and are contagious, shedding SARS-CoV-2 viral particles that can be detected on their smartphone screens. The Ct value of RT-PCR results from nasopharyngeal samples may present a level of variability that is intrinsic to the nature of the technique, the sampling process, and

because the results are not standardised to an internal reference (*Dahdouh et al., 2021*). However, considering these inherent limitations, correlations have been found between nasopharyngeal samples with low RT-PCR Ct value results and the level of infectivity assessed by the capacity to propagate the SARS-CoV-2 virus in vitro from such samples (*Bullard et al., 2020*; *Jefferson et al., 2020*; *Sonnleitner et al., 2021*). Our results add support to the possibility that patients with low Ct RT-PCR results from nasopharyngeal samples present a high viral load and are actively shedding SARS-Cov-2, which can be detected on the screen of smartphones.

Of all the *PoST*-positive samples, most of which had low nasopharyngeal RT-PCR Ct results, 76% of the pilot and 35% in the validation study were asymptomatic or pre-symptomatic at the moment of testing (*Figure 1—figure supplement 1E*, *Source data 1*, *2*). Suggesting that *PoST* is effective in identifying COVID-19-positive individuals regardless of their symptoms.

SARS-CoV-2 variant identification is required to further enhance epidemiological surveillance to contain the spreading of variants of concern that are more pathogenic or infective, to contain and isolate their dissemination. We successfully identified one *PoST* sample that presented all three polymorphisms present in Beta (B.1.351) and Gamma (P.1) variants. The rest of the tests did not unequivocally show which SARS-CoV-2 variant was present in the sample but was enough to conclude the nature of the variant present in the sample. Even though optimisation will be required to enhance SARS-CoV-2 variant identification in *PoST* samples, our results are encouraging, and could make *PoST* a viable option to screen for variants of concern. In this scenario, and although the output of results would take longer, PCR and sequencing will likely enhance the performance of variant detection in *PoST* samples.

Regarding nasopharyngeal RT-PCR results with a high Ct value (i.e., lower-viral load), these are probably in transition, and either starting or ending a COVID-19 infection. Therefore, they are less likely to shed SARS-CoV-2 virions that can be detected by *PoST*. This transition is the reason why periodic regular testing is required to identify those infected individuals once they enter a contagious phase.

To a lesser extent and against the overall trend we observe for *PoST* detection, *PoST*-positive results were identified in this high Ct value nasopharyngeal RT-PCR results group. Given the unstable nature of the SARS-CoV-2 virus on surfaces (*Goldman, 2020*), and the pervasive activity of RNases that are ubiquitously present in the environment (*Probst et al., 2006*), it is unlikely that these positive *PoST* results are due to the detection of long-lasting virus or RNA on the phone screen surface. Hence, we could exclude the possibility that these *PoST*-positive results are the consequence of detecting RNA from when individuals were passing through a moment of higher COVID-19 infection. Therefore, one could speculate that these results are the consequence of suboptimal nasopharyngeal sampling, which would explain the low amount of virus in the sample and hence, a high Ct value. Alternatively, it is plausible that these samples belong to individuals with a low viral load that are still actively contagious, which is against the observed trend (*Bullard et al., 2020*; *Jefferson et al., 2020*; *Sonnleitner et al., 2021*), but cannot be excluded as a possibility. It will be interesting to perform a follow-up study on these individuals as they could be part of a group with a higher capacity to shed SARS-CoV-2 along their infectious cycle, potentially explaining the COVID-19 superspreading capacity observed in some people (*Lewis, 2021*).

The distribution profile of the nasopharyngeal RT-PCR Ct test results in our study was different in the pilot and validation cohorts. The pilot Ct results presented an apparent bimodal distribution with low and high Ct value populations (*Figure 1B*). On the other hand, in the validation study, the distribution was unimodal and most samples tended towards low RT-PCR Ct results (*Figure 1D*). We speculate that this stark difference could be due to an inherent characteristic of these cohorts related to the positivity rates in the population of Santiago at the time of sampling. In fact, the distribution profile of the pilot nasopharyngeal Ct values is similar to that corresponding to samples taken when the pilot sampling was performed (*Figure 1—figure supplement 2G–H*). At the time the pilot study was performed, Santiago, Chile, presented a low average positivity rate (3.8% ± 1.0, *Chilean Department of Health, 2021*), and was recovering from the first COVID-19 wave. Likewise, the distribution profile of the validation cohort nasopharyngeal Ct results was similar to when Santiago was passing through the middle of first the COVID-19 wave, at a high average positivity rate (9.5–26.4%, *Figure 1—figure supplement 2B–D*). This observation suggests that the RT-PCR Ct value results distribution could show specific profile signatures that express the epidemiological

status of the population. Confirming this observation will further require a deeper analysis of this kind of dataset.

## Advantages and limitations of the *PoST* method

### Cost effective

One of the main advantages of *PoST* is its low cost compared to clinical nasopharyngeal tests. Excluding staff and premises, the net cost of a nasopharyngeal test in Chile ranges between 20 and 35 USD, which includes sample tube, transport media, swab, RNA extraction kit, and RT-PCR kit. Because *PoST* does not use clinical-grade consumables and reagents, its net cost for the same items ranges between 2 and 3 USD when it is the result of a single sample tested RT-PCR reaction. This net cost can go below $1 USD when *PoST* samples are pooled and groups of 5 and a single RT-PCR is performed, which we have shown to produce reliable results for *PoST* samples (data not shown).

Besides *PoST* not using clinical-grade consumables and reagents, one key optimisation is that *PoST* does not require an RNA extraction step, which reduces the cost by approximately $7 USD. Furthermore, instead of purchasing an RT-PCR kit specifically designed for SARS-CoV-2 detection, we assembled a combination of off-the-shelf probe/primers to detect the SARS-CoV-2 gene 'N' together with a generic RT-PCR kit. All together this optimisation enables a 10-fold price reduction of the net cost of *PoST* compared to a regular nasopharyngeal RT-PCT test.

### High sensitivity and specificity

The *PoST* method has a similar sensitivity and specificity, compared to antigen lateral flow devices, which are extensively used for routine testing (*Dinnes et al., 2021*; *Jääskeläinen et al., 2021*; *Wagenhäuser et al., 2021*). Clinical-grade diagnostic RT-PCR kits include the detection of three SARS-CoV-2 genes, plus a human positive control, either multiplexed or individually detected. The specificity reached by *PoST* in this study was achieved only when detecting the 'N' SARS-CoV-2 gene. This could explain why PoST can miss identifying some positive nasopharyngeal RT-PCR samples in the low Ct range. A study adding the detection of two or three SARS-CoV-2 genes in the *PoST* protocol to assess whether a higher sensitivity is achieved would enable us to calculate whether this trade-off is enough to justify increasing the net cost of the *PoST* assay. Especially considering that the sensitivity described in this study is already high enough to detect COVID-19-positive individuals to affect limiting the transmission of the SARS-CoV-2 virus (*Kennedy-Shaffer et al., 2021*; *Larremore et al., 2021*; *Mina et al., 2020*).

### Sampling speed and result turnaround of *PoST*

One other advantage of the *PoST* method is that the sampling process takes around a minute at most and does not require a particular setup besides having a sterile swab and sample tube. To maximise the speed and minimise errors in the *PoST* processing of samples, we developed a barcode-based smartphone application (unpublished), which enables the efficient tracking of the samples through the testing pipeline, ending with the delivery of results to the tested individuals' smartphone via SMS. Moreover, pooling samples can reduce the number of RT-PCR required to process, further decreasing the time to deliver results, when the positivity rate is below 5% to justify pooling. With all this in place, a minimal non-automatised laboratory setup with one 96-well plate real-time PCR machine has the potential to deliver 940 test results per day, when processed by two technicians, and working on two rounds of 470 samples.

Altogether, this enables an efficient turnaround of results such that after the samples arrive at the lab, the results of 940 can be delivered in approximately 5.5 hr, if no SARS-CoV-2-positive samples are found, and 6.5 hr if positive individual samples are to be identified from pools. These times consider two technicians processing the samples to feed one 96-well plate real-time PCR machine. Therefore, under these conditions, results for 940 tests could be provided within the same day of sampling, which is ideal to isolate contagious cases, and effectively curb the spreading of COVID-19.

### *PoST* as an alternative self-testing method

Due to its high sensitivity, specificity, and rapid result turnaround time, lateral flow device antigen testing has become widely used to screen for COVID-19 cases operated by trained staff

(*Pavelka et al., 2021*) and self-administered (*Riley et al., 2021*). Because this method uses highly invasive nasopharyngeal swabbing, trained operators are the preferred option to deliver accurate and reliable results from these tests. From this point of view, the *PoST* method offers a valuable alternative for accurate self-test results as it is much easier and more reliable to effectively swab the screen of a smartphone than performing a self-administered nasopharyngeal test. This, together with the fact that SARS-CoV-2 variants can be identified in *PoST* samples, gives this method further advantages compared to lateral flow antigen testing.

## Penetration of smartphone use in the population
One aspect that is important to mention is that although it is estimated that there are around 3.8 billion smartphones in the world and their penetration is very high among the young and adult population, their global distribution is not equitable (*Turner, 2021*). While penetration is almost total among adults in developed countries, in countries such as India or Bangladesh it does not exceed 33% of the population (*Berenguer et al., 2016*). On the other hand, among senior citizens, who are precisely a vulnerable population, even in developed countries some do not use a telephone or have older devices.

Overall, this study was aimed to validate *PoST* as method to identify COVID-19-infected cases by using the smartphone as a proxy from which traces of the SARS-CoV-2 virus of the owner can be detected. We propose that this highly sensitive, non-invasive, and cost-effective method could well be used for mass testing and help to contain the spreading of other airborne contagious diseases and outbreaks when tackling future epidemics. This could be particularly useful as an early warning system for rapid detection of respiratory pathogens in public health efforts to contain local outbreaks to prevent further escalating to other areas.

# Materials and methods

## Smartphone screen swab sampling, sample processing, and RT-PCR
Informed consent and consent to publish was obtained from all the individuals that participated in this study before performing the sampling process. Dacron swabs were briefly dipped in Weise medium (Merck, 1.09468.0100) and then used to swab the bottom half of mobile phone screens by a member of our team as shown in *Figure 1A*. Swabs were then introduced in sterile conical tubes containing Weise medium and briefly hand spun. Samples were processed for RT-PCR within 8 hr.

Aliquots of swab samples were incubated at 70°C for 10 min as previously described (*Miranda et al., 2020*). Samples were left to cool at room temperature and 3.3-µl aliquots were used for RT-PCR using Promega GoTaq Probe 1-Step RT-qPCR system (A6121) according to the manufacturer instructions and supplemented with SARS-CoV-2 N2 probes and primers (IDT#10006606) on an Illumina Eco Real-Time PCR System. We considered a test as positive if the PCR amplification obtained followed the expected standard sigmoidal kinetics of amplification.

## Clinical sampling and RT-PCR
For each patient, nasopharyngeal swabs were collected using standard technique, as recommended by the manufacturer (AllplexTM 2019-nCoV Assay insert) and transported to the laboratory in viral transport medium (prepared according to the standard operating procedure of the CDC, USA). For all samples, RNA extraction was performed using STARmag kit (Seegene, Korea), and following the manufacturer's instructions. For the analysis of the pilot cohort samples, target gene amplification of SARS-Cov-2 was performed using the AllplexTM 2019-nCoV Assay kit (Seegene, Korea), according to the manufacturer's procedure. The RT-qPCR preparation was carried out in the Starlet equipment (Hamilton, USA, distributed by Seegene) and the qPCR amplification in a CFX-96 thermal cycler (Bio-Rad, USA). The AllplexTM 2019-nCoV Assay kit detects three viral genes (N, RdRP, and E). We considered a test as positive if PCR amplification was obtained with N gene and RdRP. If only the E gene was amplified, the test was considered presumably positive, thus requiring repetition by another extraction instrument (MagNAPure Compact System, Roche).

## Identification of SARS-CoV-2 variants

The detection of different SARS-CoV-2 single nucleotide and deletion variants was performed using the Applied Biosystems TaqMan SARS-CoV-2 Mutation Panel kit (ThermoFisher) according to the conditions recommended by the manufacturer, on the AriaMx Real-time PCR system (Agilent Technologies) thermal cycler for fluorescence detection on VIC (reference sequence) and FAM channels (mutation sequence), and the AccuPower SARS-CoV-2 Variants ID Real-Time RT-PCR kit (Bioneer) according to manufacturer conditions, on the Exicycler 96 V4 Real Time thermal cycler (Bioneer) to detect fluorescence on the TET, TexasRed, FAM, TAMRA, and Cyanine5 channels.

## Acknowledgements

We are very grateful for the help from the following colleagues: Rachel Aguilar Montes, Prof. Miguel Allende, José Aramburó, Mauricio Daza, Dr Carmen Feijóo, Camila Gonzalez Badilla, Itziar Linazaroso, Dr Jaime Mañalich, Dr Gonzalo Mena, Emiliana Montes, Dr Gareth Powell, Rodrigo Prieto, Esteban Quiñones, Dr Cristian Undurraga, Whitmy Valdés Araneda, Dr Carlos Vega, Mr Timothy Best, Dr Kathryn Harris, Dr Julianne Brown and Healthcare Scientists at Great Ormond Street Hospital Microbiology Department RY was supported by a Moorfields Eye Charity Career Development Award and a Springboard Award. LQ was partially supported by ANID-Covid0789. FER-L and AMS were supported by ANID-Covid1038. FER-L was supported by Fondecyt 1211841.

## Additional information

### Competing interests

Rodrigo M Young, Andres Barriga-Fehrman, Carlos Abogabir: co-founder of Diagnosis Biotech. The other authors declare that no competing interests exist.

### Funding

| Funder | Grant reference number | Author |
| --- | --- | --- |
| Moorfields Eye Hospital NHS Foundation Trust | 001155 | Rodrigo M Young |
| Moorfields Eye Hospital NHS Foundation Trust | GR001210 | Rodrigo M Young |
| Agencia Nacional de Investigación e Innovación | ANID-Covid0789 | Luis A Quiñones |
| Agencia Nacional de Investigación e Innovación | ANID-Covid1038 | Ana M Sandino Felipe Reyes-Lopez |
| FONDECYT | 1211841 | Felipe Reyes-Lopez |

The funders had no role in study design, data collection and interpretation, or the decision to submit the work for publication.

### Author contributions

Rodrigo M Young, Conceptualization, Data curation, Formal analysis, Supervision, Validation, Investigation, Methodology, Writing - original draft, Project administration, Writing - review and editing, R. Y. conceived the Phone Screen Testing method; Camila J Solis, Data curation, Formal analysis, Supervision, Validation, Investigation, Project administration, Writing - review and editing; Andres Barriga-Fehrman, Conceptualization, Formal analysis, Methodology, Writing - original draft, Project administration, Writing - review and editing; Carlos Abogabir, Formal analysis, Methodology, Project administration; Alvaro R Thadani, Mariana Labarca, Eva Bustamante, Antonia G Sarda, Francisca Sepulveda, Nadia Pozas, Leslie C Cerpa, María A Lavanderos, Nelson M Varela, Alvaro Santibañez, Investigation; Cecilia V Tapia, Investigation, Methodology; Ana M Sandino, Investigation, Methodology, Project administration; Felipe Reyes-Lopez, Investigation, Methodology, Writing - original draft, Project administration; Garth Dixon, Investigation, Methodology, Writing - original draft, Writing -

review and editing; Luis A Quiñones, Supervision, Funding acquisition, Investigation, Methodology, Project administration

## Author ORCIDs
Rodrigo M Young (ID) https://orcid.org/0000-0001-5765-197X
Camila J Solis (ID) https://orcid.org/0000-0002-7195-523X
Alvaro R Thadani (ID) https://orcid.org/0000-0002-2541-1709
Mariana Labarca (ID) https://orcid.org/0000-0001-6455-3569
Eva Bustamante (ID) https://orcid.org/0000-0002-3125-8608
Cecilia V Tapia (ID) https://orcid.org/0000-0001-6234-2100
Francisca Sepulveda (ID) https://orcid.org/0000-0002-8144-0096
Leslie C Cerpa (ID) https://orcid.org/0000-0002-9525-389X
María A Lavanderos (ID) https://orcid.org/0000-0001-6167-0508
Nelson M Varela (ID) https://orcid.org/0000-0002-5229-3007
Alvaro Santibañez (ID) http://orcid.org/0000-0001-9330-2961
Ana M Sandino (ID) https://orcid.org/0000-0002-3862-3743
Felipe Reyes-Lopez (ID) https://orcid.org/0000-0002-5001-457X
Garth Dixon (ID) https://orcid.org/0000-0001-8165-3094
Luis A Quiñones (ID) https://orcid.org/0000-0002-7967-5320

## Ethics
Human subjects: Informed consent and consent to publish was obtained from all the individuals that participated in this study before performing the sampling process. This has been made explicit in the materials and methods section of the article. Ethical approval was obtained by the Ethics and Scientific Committee of Clinica Davila (Santiago, Chile) under the approval titled: "Identificación de marcadores de riesgo asociados a la severidad del Covid-19 en el microbioma respiratorio".

## Decision letter and Author response
Decision letter https://doi.org/10.7554/eLife.70333.sa1
Author response https://doi.org/10.7554/eLife.70333.sa2

# Additional files
## Supplementary files
- Source data 1. Tab 1 including all data generated by the 2020 cohort study.
- Source data 2. Tab 2 including all data generated by the 2020 cohort study.
- Source data 3. Tab 1 including all data generated by the 2020 cohort study.
- Transparent reporting form

## Data availability
All the data used generated by this study was provided in the uploaded manuscript and source files.

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
