## [Decision Letter]

[Editors’ note: the authors submitted for reconsideration following the decision after peer review. What follows is the decision letter after the first round of review.]

Thank you for submitting your work entitled "Phone Screen Testing of SARS-CoV-2 complemented by a computer-based strategy to effectively tackle the COVID-19 pandemic" for consideration by *eLife*. Your article has been reviewed by 3 peer reviewers, one of whom is a member of our Board of Reviewing Editors, and the evaluation has been overseen by a Senior Editor. The following individuals involved in review of your submission have agreed to reveal their identity: Jonathan Dushoff (Reviewer #2); Ben S Cooper (Reviewer #3).

We are sorry to say that, after consultation with the reviewers, we have decided that your work will not be considered further for publication by *eLife*.

All reviewers agreed that the testing of mobile phone screens is highly novel and this study demonstrates that it could be potentially useful to expedite more widespread testing. The major reservations related to the mathematical modeling and the lack of specific consideration about implementation issues. Reviewer two pointed out the these included models are "neither necessary or sufficient to evaluate the possible usefulness of this testing method." It would be necessary to describe the specifics of a screening plan based on smartphone testing how it might interact with other interventions, or possible disadvantages due to displacing other tests that may have higher sensitivity or specificity. Testing smartphones at scale requires a plan, and an assessment of costs, including opportunity costs.

The modeling while robust, does not account for these real world issues and is not specifically calibrated to any specific COVID epidemiology data or parameters. Moreover, the effects in the model are mediated mostly by the quarantine associated with testing, whereas the greatest utility of this testing approach would require linkage to contact tracing.

A final comment on the cell phone testing strategy was that the sample size is small, with only 51 SARS-CoV-2 individuals (by nasopharyngeal swab). Overall, this presents as an extremely promising pilot study which needs further validation in a larger cohort.

*Reviewer #1:*

Strengths of the study are:

– A novel and potentially extremely useful approach to testing;

– Exciting data showing that cell phone swabbing can clearly be used to detect most people who remain contagious;

– The key variable for this approach is turnaround time of the assay and this is addressed in the mathematical model.

Areas for potential improvement are:

– The model has not been linked to real epidemic data such that estimates of the daily percentage of people who must be tested to achieve epidemic control are likely not reliable.

– The model links testing to quarantine which is compatible with current clinical and public health practice even if testing is generally catered to people who are at risk for infection based on symptoms or contacts rather than mass screening. In that sense, the model is really capturing the effect of mass testing and quarantine. The more realistic implementation would be use of this approach in a focused test and trace model.

– In general, a more detailed and realistic model is needed to assess the degree of implementation required in order to flatten or eliminate subsequent epidemic curves.

My opinion is that the assay portion of the paper alone would do well as a short report. However, the modeling lacks sufficient link to data and details of an implementation plan to truly inform public health policy or practice.

*Reviewer #2:*

The technology proposed is interesting and may well have potential.

The authors argue for smartphone testing as a component of a COVID screening strategy. They have done a small pilot study that shows promising results. They add two sensible mathematical models which contain a lot of assumptions, at least some of which should be better justified. Crucially, these mathematical models are neither necessary or sufficient to evaluate the possible usefulness of this testing method. The authors do not discuss what a screening plan based on smartphone testing might look like, how it might interact with other interventions, or possible disadvantages due to displacing other tests that may have higher sensitivity or specificity.

The pilot study is very small. The results are certainly promising, but should not be over-interpreted. The authors do not say if the k-means based partitioning was pre-planned.

The authors dismiss their method's low sensitivity for samples with low viral loads too lightly: in particular, the claim that "only" individuals with Ct<30 expel culturable virions is not supported by the cited studies. Even less is known about "infectiousness". The authors also state that individuals early in infection are not likely to be infectious without discussing the fact that these people are likely to become infectious in reality, but not in the authors' model world, where they are not infected.

Testing smartphones at scale requires a plan, and an assessment of costs, including opportunity costs. The authors seem to assume that everyone in the population can be tested this way, either at random in the ODE model, or sequentially in the ABM. They do not discuss questions of access or of equity (which populations don't have smartphones?). The authors discuss the difficulty of nasopharyngeal testing (L60) but could provide more details about the relative advantages of smartphone testing.

The presented models are not a substitute for a clear plan or for a larger confirmatory study. There is nothing specific in the models to smartphone-based surveillance. In particular, the models do not incorporate the putative sharp difference in sensitivity to high vs. low viral loads, nor take into account how viral loads might vary through time in an infection. There are many models available to do what the authors' models do, but they are not discussed. The authors' models may well have new features, but absent a relevant literature review, the reader cannot judge.

The authors should avoid post hoc analysis pathways (like using k-means to divide into two clusters). It's probably OK to report that 26/28 high or medium-load samples were positive, but this proportion should be reported with confidence intervals and (unless the authors had a clearly written plan beforehand to not do this) the overall proportion should be reported with confidence intervals as well.

It's pretty well known that better surveillance techniques would be better. If the authors want to do their own models, they need a clear reason (one is suggested above), and they need to situate their models in the universe of similar models already available.

What is needed to make this paper convincing is more data (or better scepticism about the existing data) and more discussion about logistics.

I don't understand the claim at L123.

Not sure if this is in the reviewer's ambit, but I dislike the explanations surrounding the conflict of interest. The authors should state the facts, they don't need to state that they wish to state them, nor to judge their own conflicts.

*Reviewer #3:*

The authors describe a novel approach for detecting individuals infected with SARS-CoV-2 which avoids the need to do nasopharyngeal swabs: swabbing smartphones and testing for the virus (with RT-PCR) in the swabs. The main advantage of swabbing phones rather than the nasopharynx, the authors argue, is that costs would be lower and testing could be faster.

The authors demonstrate feasibility of such an approach by applying their smartphone screening protocol to 540 individuals, 51 of whom are positive SARS-CoV-2 (by RT-PCR) from nasopharyngeal swabs. They demonstrate remarkable concordance between the nasopharyngeal and smartphone screening results. They then perform two model-based analyses, first using a simple deterministic model with homogeneous mixing, and second using an idealised individual based model (with transmission occurring on a 100x100 grid), and show that under the model assumptions daily smartphone testing of 10% of all individuals would be able to suppress transmission of the virus provided screening results are acted on between 0 and 3 days after the swabs are taken.

Strengths:

A major strength of the work is its novelty: I am not aware of other attempts to detect SARS-CoV-2 infection routinely by screening smartphones, and if the very high concordance with nasopharyngeal swabbing achieved here could be demonstrated in a larger and better-characterised cohort the results could have considerable significance given the lower costs and greater ease with which smartphones can be screened (it could, for example, be mechanised reducing costs further).

Secondly, quite apart from whether the findings do have potential to be used in routine screening as the authors suggest, the finding concerning the agreement between results from screening phones and nasopharynx has infection control implications which, though not mentioned by the authors, may be of potential importance in some contexts.

Weaknesses:

The sample size is small, with only 51 SARS-CoV-2 individuals (by nasopharyngeal swab) so this must be considered a pilot study rather than a large-scale evaluation of the approach. Clearly, if the approach is to gain any traction a much larger study is needed.

The claimed advantages over conventional nasopharyngeal screening are lower costs and faster turnaround times, but no attempt to quantify these are given. This information seems crucial for comparison with other testing approaches (e.g. PCR-based tests of nasopharyngeal swabs and lateral flow tests which are now widely used).

While the modelling seems to be done with competence, conclusions are based on idealised assumptions. For example, models assume once isolated infected individuals do not contribute to further transmission. At least in settings where isolation occurs at home (which is most parts of the world) this is not a realistic assumption. Other aspects of the models also lack biological realism (for example, the assumption that infectivity does not change over the course of an infection and exponential sojourn times in model compartments). For these reasons, the claim that "iterative testing over 10% of the population [daily] could effectively suppress the spread of COVID-19" applies only to the hypothetical idealised world of the model, and the work tells us little about how such at an approach would work in practice (accounting for the fact that isolation will not stop within-household transmission and that isolation will be imperfect etc).

Comments for the authors:

1. Some of the terminology used is non-standard. For example "quarantine" traditionally applies to isolation of those considered at risk of being infected but who have not yet been shown to be infected. "isolation" would be a better term for what the authors are describing.

2. I don't think the comparison with influenza in 1918/19 and talk of the "third wave" (as a universal occurrence) is helpful – it's a very different pathogen, and different countries may experience different numbers of "waves" due to the timing of interventions, emerging variants and seasonal factors.

3. To better make the case for the advantages of the approach over nasopharyngeal screening, some numbers are needed: costs, turnaround times etc.

4. Very little information is given on how those screened are selected. This information would strengthen the paper.

5. To make the case for the value of the approach in a specific population information on smartphone ownership in different age groups is needed.

6. Another potential advantage over nasopharyngeal swabbing is the lack of physical discomfort. This could lead to greater acceptance of the approach. In future work, it would be interesting to explore this.

7. Clearly, if the remarkable findings reported are to translate into a real-world intervention a much larger study with a better characterised population is needed. In future work the authors could also consider exploring RT-PCR versus lateral flow testing of the phone swabs.

8. While the models reported are adequate for a rough back-of-the envelope assessment, a more thorough model-based assessment would require models that have a stronger biological basis (i.e. more realistic assumptions about how infectivity changes over time, see for example, Grassly et al. Lancet Inf Dis 2020), and also more realistic assumptions about isolation (i.e. adherence and transmission in isolation settings).

9. In line with the approach in Grassly et al., I think it would be more informative to report modelling results as % reductions in transmission, as baseline transmissibility varies widely between settings (due to variation in climate, behaviours, pathogen lineage etc).

10. Very few results are reported from the agent-based model (only the kurtosis plots). It wasn't clear to me why this model wasn't explored more fully.

[Editors’ note: further revisions were suggested prior to acceptance, as described below.]

Thank you for submitting your article "Smartphone Screen Testing, a novel pre-diagnostic method to identify SARS-CoV-2 infectious individuals" for consideration by *eLife*. Your article has been reviewed by Joshua T Schiffer as the Reviewing Editor and Reviewer #1 and Jos van der Meer as the Senior Editor.

The Reviewing Editor has drafted this to help you prepare a revised submission.

*Essential revisions:*

1) Please provide a more balanced comparison of cell phone screening versus lateral flow antigen testing.

*Reviewer #1:*

This manuscript presents cell phone swabbing as a potentially useful method for SARS-CoV-2 diagnosis.

The revised version of this paper does a nice job of adding more samples and clarifying certain analyses. The removal of the mathematical modeling strengthens the manuscript. The paper demonstrates that self-phone sampling for SARS-CoV-2 appears to be a promising diagnostic approach, worthy of more study.

One criticism is that the advantages relative to lateral flow antigen testing are overstated. Antigen tests offer similar sensitivity and specificity with more rapid turnaround time. This should be mentioned.

---

## [Author Response]

[Editors’ note: the authors resubmitted a revised version of the paper for consideration. What follows is the authors’ response to the first round of review.]

All reviewers agreed that the testing of mobile phone screens is highly novel and this study demonstrates that it could be potentially useful to expedite more widespread testing. The major reservations related to the mathematical modeling and the lack of specific consideration about implementation issues. Reviewer two pointed out the these included models are "neither necessary or sufficient to evaluate the possible usefulness of this testing method." It would be necessary to describe the specifics of a screening plan based on smartphone testing how it might interact with other interventions, or possible disadvantages due to displacing other tests that may have higher sensitivity or specificity. Testing smartphones at scale requires a plan, and an assessment of costs, including opportunity costs.The modeling while robust, does not account for these real world issues and is not specifically calibrated to any specific COVID epidemiology data or parameters. Moreover, the effects in the model are mediated mostly by the quarantine associated with testing, whereas the greatest utility of this testing approach would require linkage to contact tracing.A final comment on the cell phone testing strategy was that the sample size is small, with only 51 SARS-CoV-2 individuals (by nasopharyngeal swab). Overall, this presents as an extremely promising pilot study which needs further validation in a larger cohort.

We thank the reviewers for their very helpful comments, all of which have been taken on board in this new improved manuscript. Besides removing the computational modelling section from the manuscript, the three main changes in the new version are:

1. In addition to the 51 nasopharyngeal SARS-CoV-2 RT-PCR positive samples, 182 new cases were added to the study.

2. We show that SARS-CoV-2 ‘variants of concern’ can be identified from the samples taken from smartphone screens.

3. We have extensively elaborated on the interpretation and limitations of our results in the Discussion section of the new manuscript.

In the following rebuttal, we have only addressed the comments of reviewers regarding the SARSCoV-2 phone screen testing method. Below, we provide a more detailed description of our responses to the reviewer comments.

Reviewer #1:[…] My opinion is that the assay portion of the paper alone would do well as a short report. However, the modeling lacks sufficient link to data and details of an implementation plan to truly inform public health policy or practice.

We have removed the modelling data from the manuscript.

Reviewer #2:[…] The pilot study is very small. The results are certainly promising, but should not be over-interpreted. The authors do not say if the k-means based partitioning was pre-planned.

We took on board this and other reviewers’ comments regarding sample number and, as mentioned above, we have increased the sample number by 182 nasopharyngeal positive RT-PCR results. Also following this and other reviewers’ suggestions we have further elaborated on many aspects in the discussion of this new version and have also removed the k-means portioning as based on the new data, we decided it is no longer necessary.

[…] The authors should avoid post hoc analysis pathways (like using k-means to divide into two clusters). It's probably OK to report that 26/28 high or medium-load samples were positive, but this proportion should be reported with confidence intervals and (unless the authors had a clearly written plan beforehand to not do this) the overall proportion should be reported with confidence intervals as well.

The k-means post hoc analysis has been removed from the manuscript.

[…] What is needed to make this paper convincing is more data (or better scepticism about the existing data) and more discussion about logistics.

We have increased the SARS-CoV-2 Rt-PCR positive sample number four times and also extensively elaborated on the implications and potential of this new method in the discussion.

I don't understand the claim at L123.

This has been removed from the current manuscript.

Not sure if this is in the reviewer's ambit, but I dislike the explanations surrounding the conflict of interest. The authors should state the facts, they don't need to state that they wish to state them, nor to judge their own conflicts.

We agree with the reviewer and have now just stated the facts.

Reviewer #3:[…] The sample size is small, with only 51 SARS-CoV-2 individuals (by nasopharyngeal swab) so this must be considered a pilot study rather than a large-scale evaluation of the approach. Clearly, if the approach is to gain any traction a much larger study is needed.

As mentioned above, we have significantly increased the sample number of SARS-CoV-2 positive individuals tested by nasal swabbing and RT-PCR. We expect this is enough for the referees to now consider this a validated method.

The claimed advantages over conventional nasopharyngeal screening are lower costs and faster turnaround times, but no attempt to quantify these are given. This information seems crucial for comparison with other testing approaches (e.g. PCR-based tests of nasopharyngeal swabs and lateral flow tests which are now widely used).

We thank the reviewer for pointing this out. Based on these suggestions we have made significant additions to the discussion which now includes arguments and quantification backing up the lowcost and quick turnaround times we claim for the phone screen testing method.

[…] Comments for the authors:1. Some of the terminology used is non-standard. For example "quarantine" traditionally applies to isolation of those considered at risk of being infected but who have not yet been shown to be infected. "isolation" would be a better term for what the authors are describing.

This comparison was removed from the introduction.

2. I don't think the comparison with influenza in 1918/19 and talk of the "third wave" (as a universal occurrence) is helpful – it's a very different pathogen, and different countries may experience different numbers of "waves" due to the timing of interventions, emerging variants and seasonal factors.

As mentioned above, this has now been included in the discussion.

3. To better make the case for the advantages of the approach over nasopharyngeal screening, some numbers are needed: costs, turnaround times etc.

As the reviewer has noticed, to maximise the number of tested individuals, there was no selection criteria and all the people who arrived at the Davila Clinic (Santiago, Chile) for a SARS-CoV-2 test were recruited for the study.

[…] 5. To make the case for the value of the approach in a specific population information on smartphone ownership in different age groups is needed.

We have included studies for geographical and age distribution of smartphone usage in the discussion.

6. Another potential advantage over nasopharyngeal swabbing is the lack of physical discomfort. This could lead to greater acceptance of the approach. In future work, it would be interesting to explore this.

We agree with the reviewer and the non-invasive advantage of the phone screen testing method was enhanced in this new version of the manuscript.

7. Clearly, if the remarkable findings reported are to translate into a real-world intervention a much larger study with a better characterised population is needed. In future work the authors could also consider exploring RT-PCR versus lateral flow testing of the phone swabs.

We agree with the reviewer and we are currently embarked on a study comparing lateral flow device antigen testing and phone screen testing results. Comparisons between both methods have also been elaborated in the discussion of this current version of the manuscript.

[Editors’ note: what follows is the authors’ response to the second round of review.]

Reviewer #1:The revised version of this paper does a nice job of adding more samples and clarifying certain analyses. The removal of the mathematical modeling strengthens the manuscript. The paper demonstrates that self-phone sampling for SARS-CoV-2 appears to be a promising diagnostic approach, worthy of more study.One criticism is that the advantages relative to lateral flow antigen testing are overstated. Antigen tests offer similar sensitivity and specificity with more rapid turnaround time. This should be mentioned.

We agree with the reviewer and when referring to lateral flow antigen testing, we have replaced:

“Lateral flow device antigen testing has become widely used to screen for COVID-19 cases operated by trained staff (Pavelka et al., 2021), and self-administered (Riley et al., 2021).”

For:

“Due to its high sensitivity, specificity, and rapid result turnaround time, lateral flow device antigen testing has become widely used to screen for COVID-19 cases operated by trained staff (Pavelka et al., 2021), and self-administered (Riley et al., 2021).”